# Statistical Test on Diffusion Model-based Anomaly Detection by Selective Inference

## Abstract

Advancements in AI image generation, particularly diffusion models, have progressed rapidly. However, the absence of an established framework for quantifying the reliability of AI-generated images hinders their use in critical decision-making tasks, such as medical image diagnosis. In this study, we address the task of detecting anomalous regions in medical images using diffusion models and propose a statistical method to quantify the reliability of the detected anomalies. The core concept of our method involves a selective inference framework, wherein statistical tests are conducted under the condition that the images are produced by a diffusion model. With our approach, the statistical significance of anomaly detection results can be quantified in the form of a $p$-value, enabling decision-making with controlled error rates, as is standard in medical practice. We demonstrate the theoretical soundness and practical effectiveness of our statistical test through numerical experiments on both synthetic and brain image datasets.

## 1 Introduction

Advances in image generation AI, such as diffusion models, have been remarkable (Song & Ermon, 2019). They can generate highly realistic and detailed images, which leads to innovations in various tasks across various fields. For example, image generation AI significantly enhances medical image diagnosis by improving accuracy and efficiency. It can generate highly detailed and enhanced images from standard medical scan images, potentially offering doctors to detect anomalies and diseases with greater precision. Furthermore, image generation AI can be used to create alternative versions of medical images to consider *what-if* scenarios. For example, it can generate virtual images of a patient when they are healthy, which allows for comparing the current actual images with the virtual healthy images, making it possible to provide a diagnosis tailored to the individual patient.

On the other hand, when using virtual images generated by AI for critical decision-making, such as medical diagnosis, it is crucial to ensure the reliability of the decisions. Given that images are generated by an AI algorithm, such as a deep learning model trained on historical data, they may inherently contain biases and errors. Therefore, treating virtual synthetic images as equivalent to real images in decision-making tasks carries the risk of biased and erroneous outcomes. When making critical decisions based on generated images, it is necessary to be able to assess their reliability by properly taking into account the fact that the images were generated by AI. However, to our knowledge, there are no studies that can quantify the reliability of decision-making based on image generation AI.

In this study, we address this challenge using the statistical hypothesis testing framework. We introduce a statistical inference framework called *selective inference (SI)*, which has gained attention over the past decade in the statistics community as a novel approach for data-driven hypotheses (Taylor & Tibshirani, 2015; Fithian et al., 2015; Lee & Taylor, 2014). In SI, statistical inference is performed based on the sampling distribution of the test statistic under the condition that the hypothesis being tested was selected based on the data. Our core idea is to formulate decision-making tasks involving generated images as statistical hypothesis testing problems, and to incorporate SI framework to accurately quantify the reliability of decisions influenced by these generated images.

As an example of decision-making tasks based on image generation AI, we focus on the problem of detecting anomalous regions in medical images (Wolleb et al., 2022; Baur et al., 2021) (see Figure 1). Initially, a diffusion model is trained exclusively on normal images during the training phase. In the

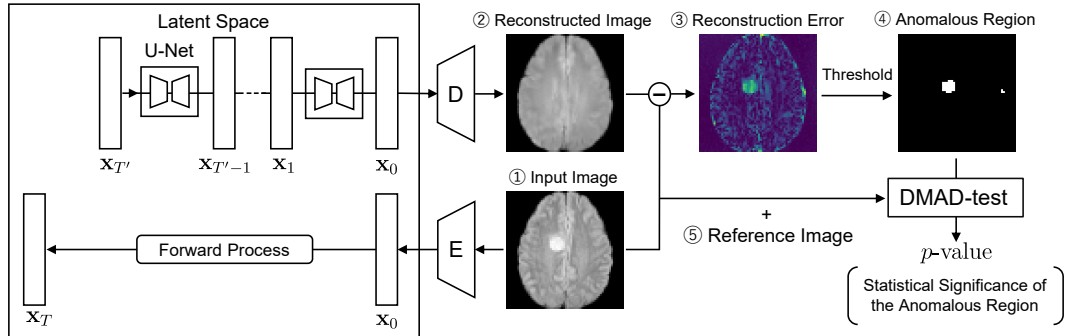

Figure 1: Schematic illustration of the anomaly detection on a brain image dataset using a diffusion model and the proposed DMAD-test. When a test image, which may contain an anomalous region, is fed into a trained diffusion model, a normal image is generated through the forward process and reverse process. By initiating the image generation from the middle of the forward process, a normal image that retains the characteristics of the input image can be generated. By comparing the input image with the normal image, the anomalous region can be identified. In this study, we propose a method called the DMAD-test, which quantifies the statistical significance of the identified anomalous regions in the form of $p$-value. The DMAD-test calculates the $p$-values by incorporating the fact that the anomalous region has been identified by the diffusion model, thus enabling unbiased decision-making (see §3 and §4 for details).

testing phase, a patient's test image is processed through this model to create a virtual normal image, against which the original is compared to identify anomalous regions. Our proposed statistical test, *the Diffusion Model-based Anomalous Region Detection Test (DMAD-test)*, quantifies the statistical reliability of detected anomalies as $p$-values. Decisions based on these $p$-values can theoretically control the false detection rate at desired significance levels (such as 0.01 or 0.05).

**Related work.** Diffusion models have been effectively utilized in anomalous region detection problems (Wolleb et al., 2022; Pinaya et al., 2022; Fontanella et al., 2023; Wyatt et al., 2022; Mousakhan et al., 2023). In this context, the *denoising diffusion probabilistic model (DDPM)* is commonly used (Ho et al., 2020; Song et al., 2022). During the training phase, a DDPM model learns the distribution of normal medical images by iteratively adding and then removing noise. In the test phase, the model attempts to reconstruct a new test image. If the image contains anomalous regions, such as tumors, the model may struggle to accurately reconstruct these regions, as it has been trained primarily on normal regions. The discrepancies between the original and the reconstructed image are then analyzed to identify and highlight anomalous regions. Other types of generative AI has also been used for anomalous region detection task (Baur et al., 2021; Chen & Konukoglu, 2018; Chow et al., 2020; Jana et al., 2022).

SI was first introduced within the context of reliability evaluation for linear model features when they were selected using a feature selection algorithm (Lee & Taylor, 2014; Lee et al., 2016; Tibshirani et al., 2016), and then extended to more complex feature selection methods (Yang et al., 2016; Suzumura et al., 2017; Hyun et al., 2018; Rügamer & Greven, 2020; Das et al., 2021). Then, SI proves valuable not only for feature selection problems but also for statistical inference across various data-driven hypotheses, including unsupervised learning tasks (Chen & Bien, 2020; Tsukurimichi et al., 2021; Tanizaki et al., 2020; Duy et al., 2022; Le Duy et al., 2024; Lee et al., 2015; Gao et al., 2022; Duy et al., 2020; Jewell et al., 2022). The fundamental idea of SI is to perform an inference conditional on the hypothesis selection event, which mitigates the selection bias issue even when the hypothesis is selected and tested using the same data. To conduct SI, it is necessary to derive the sampling distribution of test statistic conditional on the hypothesis selection event. To the best of our knowledge, SI was applied to statistical inferences on several deep learning models (Duy et al., 2022; Miwa et al., 2023; Shiraishi et al., 2024b; Miwa et al., 2024), but none of them works on image generation by diffusion models.

**Contributions.** Our main contributions in this study are summarized as follows. Our first contribution is the introduction of a statistical testing framework for quantifying reliability in decision-

making based on images generated by diffusion models. The second contribution is the implementation of SI for diffusion models, which requires the calculation of the sampling distribution conditional on the diffusion model, necessitating the development of non-trivial computational methodology. The third contribution is to theoretically guarantee the performance of the proposed DMAD-test and demonstrate its performance through numerical experiments and applications in brain imaging diagnostics. The code is available as supplementary material.

## 2   DIFFUSION MODELS

In this section, we briefly explain the diffusion model employed in this study. Given a test image which possibly contain anomalous regions, a denoising diffusion model (Ho et al., 2020; Song et al., 2022) is used to generate the corresponding normal image. The reconstruction process consists of two processes called *forward process (or diffusion process)* and *reverse process*.

In the forward process, noise is sequentially added to the test image so that it converges to a standard Gaussian distribution $\mathcal{N}(\mathbf{0}, I)$. Let $\mathbf{x}$ be an image represented as a vector with each element corresponding to a pixel value. Given an original test image $\mathbf{x}_0$, noisy images $\mathbf{x}_1, \mathbf{x}_2, \ldots, \mathbf{x}_T$ are sequentially generated, where $T$ is the number of noise addition steps. We consider the distribution of the original and noisy test images, which is denoted by $q(\mathbf{x})$, and approximate the distribution by a parametric model $p_\theta(\mathbf{x})$ with $\theta$ being the parameters. Using a sequence of noise scheduling parameters $0 < \beta_1 < \beta_2, < \cdots < \beta_T < 1$, the forward process is written as

$$q(\mathbf{x}_{1:T}|\mathbf{x}_0) := \prod_{t=1}^{T} q(\mathbf{x}_t|\mathbf{x}_{t-1}), \quad \text{where} \quad q(\mathbf{x}_t|\mathbf{x}_{t-1}) := \mathcal{N}(\sqrt{1-\beta_t}\mathbf{x}_{t-1}, \beta_t I).$$

By the reproducibility of the Gaussian distribution, $\mathbf{x}_t$ can be rewritten by a linear combination of $\mathbf{x}_0$ and $\epsilon$, i.e.,

$$\mathbf{x}_t = \sqrt{\alpha_t}\mathbf{x}_0 + \sqrt{1-\alpha_t}\epsilon, \quad \text{with} \quad \epsilon \sim \mathcal{N}(\mathbf{0}, I), \tag{1}$$

where $\alpha_t = \prod_{s=1}^{t}(1-\beta_s)$.

In the reverse process, a parametric model in the form of $p_\theta(\mathbf{x}_{t-1}|\mathbf{x}_t) = \mathcal{N}(\mathbf{x}_{t-1}; \mu_\theta(\mathbf{x}_t, t), \beta_t I)$ is employed, where $\mu_\theta(\mathbf{x}_t, t)$ is obtained by using the predicted noise component $\epsilon_\theta^{(t)}(\mathbf{x}_t)$. Typically, a U-Net is used as the model architecture for $\epsilon_\theta^{(t)}(\mathbf{x}_t)$. In DDPM (Ho et al., 2020), the loss function for training the noise component is simply written as $||\epsilon_\theta^{(t)}(\mathbf{x}_t) - \epsilon_t||_2^2$. Based on (1), given a noisy image $\mathbf{x}_t$ after $t$ steps, the reconstruction of the image in the previous step $\mathbf{x}_{t-1}$ is obtained as

$$\mathbf{x}_{t-1} = \sqrt{\alpha_{t-1}} \cdot f_\theta^{(t)}(\mathbf{x}_t) + \sqrt{1-\alpha_{t-1}-\sigma_t^2} \cdot \epsilon_\theta^{(t)}(\mathbf{x}_t) + \sigma_t\epsilon_t, \tag{2}$$

where

$$f_\theta^{(t)}(\mathbf{x}_t) := (\mathbf{x}_t - \sqrt{1-\alpha_t} \cdot \epsilon_\theta^{(t)}(\mathbf{x}_t))/\sqrt{\alpha_t}, \tag{3}$$

and

$$\sigma_t = \eta\sqrt{(1-\alpha_{t-1})/(1-\alpha_t)}\sqrt{1-\alpha_t/\alpha_{t-1}}. \tag{4}$$

Here, $\eta$ is a hyperparameter that controls the randomness in the reverse process. By setting $\eta = 1$, we can create new images by stochastic sampling. On the other hand, if we set $\eta = 0$, deterministic sampling is used for image generation. By recursively sampling as in (2), we can obtain a reconstructed image of the original input $\mathbf{x}_0$.

In practice, the reverse process starts from $\mathbf{x}_{T'}$ with $T' < T$. Namely, we reconstruct the original input image not from the completely noisy one, but from a one which still contains individual information of the original input image. The smaller $T'$ ensures that the reconstructed image preserves fine details of the input image. Conversely, the larger $T'$ results in the retention of only large scale features, thereby converting more of the anomalous regions into normal regions (Ho et al., 2020; Mousakhan et al., 2023). Therefore, $T'$ should be set to balance the feature retention of the input image and the conversion of the anomalous region to the normal region. Note that setting $T'$ smaller than $T$ has advantages in terms of computational cost. For the purpose of reducing computational cost, various methods have been proposed. For example, one way is to sample while skipping portions of the sampling trajectory (see Appendix A). The image reconstruction scheme by DDPM is summarized in Algorithm 1.

---

**Algorithm 1** Reconstruction Process

---

**Require:** Input image $\mathbf{x}$
1: $\mathbf{x}_{T'} \leftarrow \sqrt{\alpha_{T'}}\mathbf{x} + \sqrt{1 - \alpha_{T'}}\epsilon$
2: **for** $t = T', \dots, 1$ **do**
3: $\quad f_\theta^{(t)}(\mathbf{x}_t) \leftarrow (\mathbf{x}_t - \sqrt{1 - \alpha_t} \cdot \epsilon_\theta^{(t)}(\mathbf{x}_t))/\sqrt{\alpha_t}$
4: $\quad \mathbf{x}_{t-1} \leftarrow \sqrt{\alpha_{t-1}} \cdot f_\theta^{(t)}(\mathbf{x}_t) + \sqrt{1 - \alpha_{t-1} - \sigma_t^2} \cdot \epsilon_\theta^{(t)}(\mathbf{x}_t) + \sigma_t \epsilon_t$
5: **end for**
**Ensure:** Reconstructed image $\mathbf{x}_0$

---

## 3 STATISTICAL TEST ON GENERATED IMAGES BY DIFFUSION MODELS

In this section, we formulate the statistical test for detecting anomalous regions using images generated by a trained DDPM model. As shown in Figure 1, anomalous region detection by diffusion models is conducted as follows. First, in the training phase, the diffusion model is trained only on normal images. Then, in the test phase, we feed a test image which might contain anomalous regions into the trained diffusion model, and reconstruct it back from a noisy image $\mathbf{x}_{T'}$ at step $T' < T$. By appropriately selecting $T'$, we can generate a normal image that retain individual characteristics of the test input image. If the image does not contain anomalous regions, the reconstructed image is expected to be similar to the original test image. On the other hand, if the image contains anomalous regions, such as tumors, the model may struggle to accurately reconstruct these regions, as it has been trained primarily on normal regions. Therefore, the anomalous regions can be detected by comparing the original test image and its reconstructed one.

**Problem formulation.** We develop a statistical test to quantify the reliability of decision-making based on images generated by diffusion models. To develop a statistical test, we interpret an image as a sum of a true signal component $\boldsymbol{\mu} \in \mathbb{R}^n$ and a noise component $\boldsymbol{\varepsilon} \in \mathbb{R}^n$. We emphasize that the noise component $\boldsymbol{\varepsilon}$ should not be confused with the noise $\epsilon$ used in the forward process. Regarding the true signal component, each pixel can have an arbitrary value without any particular assumption or constraint. On the other hand, regarding the noise component, it is assumed to follow a Gaussian distribution, and their covariance matrix is estimated using normal data different from that used for the training of the diffusion model, which is the standard setting of SI. Namely, an image with $n$ pixels can be represented as an $n$-dimensional random vector

$$\boldsymbol{X} = (X_1, X_2, \dots, X_n)^\top = \boldsymbol{\mu} + \boldsymbol{\varepsilon}, \quad \boldsymbol{\varepsilon} \sim \mathcal{N}(\boldsymbol{0}, \Sigma),$$

where $\boldsymbol{\mu} \in \mathbb{R}^n$ is the unknown true signal vector and $\Sigma$ is the covariance matrix. In the following, we use capital $\boldsymbol{X}$ to emphasize that an image is considered as a random vector, while the observed image is denoted as $\boldsymbol{X}_{\mathrm{obs}}$.

Let us denote the reconstruction process of the trained diffusion model in Algorithm 1 as the mapping from an input image to the reconstructed image $\mathcal{D} : \mathbb{R}^n \ni \boldsymbol{X} \to \mathcal{D}(\boldsymbol{X}) \in \mathbb{R}^n$. The difference between the input image $\boldsymbol{X}$ and the reconstructed image $\mathcal{D}(\boldsymbol{X})$ indicates the reconstruction error. When identifying anomalous regions based on reconstruction error, it is useful to apply some filter to remove the influence of pixel-wise noise. In this study, we simply used an averaging filter. Let us denote the averaging filter as $\mathcal{F} : \mathbb{R}^n \to \mathbb{R}^n$. Then, the process of obtaining the (filtered) reconstruction error is written as

$$E : \mathbb{R}^n \ni \boldsymbol{X} \mapsto |\mathcal{F}(\boldsymbol{X} - \mathcal{D}(\boldsymbol{X}))| \in \mathbb{R}^n,$$

where absolute value is taken pixel-wise. Anomalous regions are then obtained by applying a threshold to the filtered reconstruction error $E_i(\boldsymbol{X})$ for each pixel $i \in [n]$. Specifically, we define the anomalous region as the set of pixels whose filtered reconstruction error is greater than a given threshold $\lambda \in (0, \infty)$, i.e.,

$$\mathcal{M}_{\boldsymbol{X}} = \{i \in [n] \mid E_i(\boldsymbol{X}) \geq \lambda\}. \tag{5}$$

**Statistical inference.** In order to quantify the statistical significance of the anomalous regions detected by using a diffusion model, we consider the concrete example of two-sample test. Note that our method can be extended to other statistical tests using various statistics. In the two-sample

test, we compare the test image and the randomly chosen reference image in the anomalous region. Let us define an $n$-dimensional reference input vector,

$$\boldsymbol{X}^{\text{ref}} = (X_1^{\text{ref}}, X_2^{\text{ref}}, \ldots, X_n^{\text{ref}})^\top = \boldsymbol{\mu}^{\text{ref}} + \boldsymbol{\varepsilon}^{\text{ref}}, \quad \boldsymbol{\varepsilon}^{\text{ref}} \sim \mathcal{N}(\boldsymbol{0}, \Sigma),$$

where $\boldsymbol{\mu}^{\text{ref}} \in \mathbb{R}^n$ is the unknown true signal vector of the reference image and the $\boldsymbol{\varepsilon}^{\text{ref}} \in \mathbb{R}^n$ is the noise component. Then, we consider the following null and alternative hypotheses:

$$\text{H}_0 \colon \frac{1}{|\mathcal{M}_{\boldsymbol{X}}|} \sum_{i \in \mathcal{M}_{\boldsymbol{X}}} \mu_i = \frac{1}{|\mathcal{M}_{\boldsymbol{X}}|} \sum_{i \in \mathcal{M}_{\boldsymbol{X}}} \mu_i^{\text{ref}} \quad \text{v.s.} \quad \text{H}_1 \colon \frac{1}{|\mathcal{M}_{\boldsymbol{X}}|} \sum_{i \in \mathcal{M}_{\boldsymbol{X}}} \mu_i \neq \frac{1}{|\mathcal{M}_{\boldsymbol{X}}|} \sum_{i \in \mathcal{M}_{\boldsymbol{X}}} \mu_i^{\text{ref}}, \quad (6)$$

where $\text{H}_0$ is the null hypothesis that the mean pixel values are the same between the test image and the reference images in the anomalous regions, while $\text{H}_1$ is the alternative hypothesis that they are different. A reasonable test statistic for the statistical test in (6) is the difference in mean pixel values between the test image and the reference image in the anomalous region $\mathcal{M}_{\boldsymbol{X}}$, i.e.,

$$T(\boldsymbol{X}, \boldsymbol{X}^{\text{ref}}) = \frac{1}{|\mathcal{M}_{\boldsymbol{X}}|} \sum_{i \in \mathcal{M}_{\boldsymbol{X}}} X_i - \frac{1}{|\mathcal{M}_{\boldsymbol{X}}|} \sum_{i \in \mathcal{M}_{\boldsymbol{X}}} X_i^{\text{ref}} = \boldsymbol{\nu}_{\mathcal{M}_{\boldsymbol{X}}}^\top \begin{pmatrix} \boldsymbol{X} \\ \boldsymbol{X}^{\text{ref}} \end{pmatrix},$$

where $\boldsymbol{\nu}_{\mathcal{M}_{\boldsymbol{X}}} \in \mathbb{R}^{2n}$ is the vector that depends on the anomalous region $\mathcal{M}_{\boldsymbol{X}}$, defined as

$$\boldsymbol{\nu}_{\mathcal{M}_{\boldsymbol{X}}} = \frac{1}{|\mathcal{M}_{\boldsymbol{X}}|} \begin{pmatrix} \boldsymbol{1}_{\mathcal{M}_{\boldsymbol{X}}}^n \\ -\boldsymbol{1}_{\mathcal{M}_{\boldsymbol{X}}}^n \end{pmatrix} \in \mathbb{R}^{2n},$$

where $\boldsymbol{1}_{\mathcal{C}}^n \in \mathbb{R}^n$ is an $n$-dimensional vector whose elements are 1 if they belong to the set $\mathcal{C}$ and 0 otherwise. If we do not account for the fact that the anomalous region is detected by a diffusion model, the distribution of the test statistic would be simply given as

$$T(\boldsymbol{X}, \boldsymbol{X}^{\text{ref}}) \sim \mathcal{N}(0, \boldsymbol{\nu}_{\mathcal{M}_{\boldsymbol{X}}}^\top \tilde{\Sigma} \boldsymbol{\nu}_{\mathcal{M}_{\boldsymbol{X}}}), \quad \text{where} \quad \tilde{\Sigma} = \begin{pmatrix} \Sigma & O_n \\ O_n & \Sigma \end{pmatrix}.$$

In this case, the $p$-values defined as

$$p_{\text{naive}} = \mathbb{P}_{\text{H}_0}\left(|T(\boldsymbol{X}, \boldsymbol{X}^{\text{ref}})| > |T(\boldsymbol{X}_{\text{obs}}, \boldsymbol{X}_{\text{obs}}^{\text{ref}})|\right),$$

would be easily computed by the normality of the test statistic distribution. However, in reality, since the anomalous region is detected by the trained diffusion model, $\boldsymbol{\nu}_{\mathcal{M}_{\boldsymbol{X}}}$ depends on the data $\boldsymbol{X}$, meaning that the sampling distribution of the test statistic is much more complicated. Therefore, if $p_{\text{naive}}$ is used for decision-making, the false detection error cannot be properly controlled.

## 4 COMPUTING SELECTIVE $p$-VALUES

In this section, we introduce selective inference (SI) framework for testing images generated by diffusion models and propose a method to perform valid hypothesis test.

### 4.1 CONDITIONAL DISTRIBUTION OF TEST STATISTICS

Due to the complexity described in the previous section, it is difficult to directly obtain the sampling distribution of $T(\boldsymbol{X}, \boldsymbol{X}^{\text{ref}})$. Then, we consider the sampling distribution of $T(\boldsymbol{X}, \boldsymbol{X}^{\text{ref}})$ conditional on the event that the anomalous region $\mathcal{M}_{\boldsymbol{X}}$ is the same as the observed anomalous region $\mathcal{M}_{\boldsymbol{X}_{\text{obs}}}$, i.e.,

$$T(\boldsymbol{X}, \boldsymbol{X}^{\text{ref}}) \mid \{\mathcal{M}_{\boldsymbol{X}} = \mathcal{M}_{\boldsymbol{X}_{\text{obs}}}\}.$$

In the context of SI, to make the characterization of the conditional sampling distribution manageable, we also incorporate conditioning on the nuisance parameter that is independent of the test statistic. As a result, the calculation of the conditional sampling distribution in SI can be reduced to a one-dimensional search problem in an $n$-dimensional data space. The nuisance parameter $\mathcal{Q}_{\boldsymbol{X}, \boldsymbol{X}^{\text{ref}}}$ is written as

$$\mathcal{Q}_{\boldsymbol{X}, \boldsymbol{X}^{\text{ref}}} = \left(I_{2n} - \frac{\tilde{\Sigma} \boldsymbol{\nu}_{\mathcal{M}_{\boldsymbol{X}}} \boldsymbol{\nu}_{\mathcal{M}_{\boldsymbol{X}}}^\top}{\boldsymbol{\nu}_{\mathcal{M}_{\boldsymbol{X}}}^\top \tilde{\Sigma} \boldsymbol{\nu}_{\mathcal{M}_{\boldsymbol{X}}}}\right) \begin{pmatrix} \boldsymbol{X} \\ \boldsymbol{X}^{\text{ref}} \end{pmatrix}.$$

The $p$-value calculated from this conditional sampling distribution is called a selective $p$-value. Specifically, the selective $p$-value is defined as

$$p_{\text{selective}} = \mathbb{P}_{\text{H}_0}\left(|T(\boldsymbol{X}, \boldsymbol{X}^{\text{ref}})| > |T(\boldsymbol{X}_{\text{obs}}, \boldsymbol{X}_{\text{obs}}^{\text{ref}})| \mid \boldsymbol{X} \in \mathcal{X}\right), \qquad (7)$$

where $\mathcal{X}$ is the conditional data space defined as

$$\mathcal{X} = \left\{ \begin{pmatrix} \boldsymbol{X} \\ \boldsymbol{X}^{\text{ref}} \end{pmatrix} \in \mathbb{R}^{2n} \middle| \mathcal{M}_{\boldsymbol{X}} = \mathcal{M}_{\boldsymbol{X}_{\text{obs}}}, \mathcal{Q}_{\boldsymbol{X}, \boldsymbol{X}^{\text{ref}}} = \mathcal{Q}_{\boldsymbol{X}_{\text{obs}}, \boldsymbol{X}_{\text{obs}}^{\text{ref}}} \right\}.$$

Due to the conditioning on the nuisance parameter $\mathcal{Q}_{\boldsymbol{X}}$, the conditional data space $\mathcal{X}$ can be rewritten as

$$\mathcal{X} = \left\{ \begin{pmatrix} \boldsymbol{X}(z) \\ \boldsymbol{X}^{\text{ref}}(z) \end{pmatrix} \in \mathbb{R}^{2n} \middle| \begin{pmatrix} \boldsymbol{X}(z) \\ \boldsymbol{X}^{\text{ref}}(z) \end{pmatrix} = \boldsymbol{a} + \boldsymbol{b}z, z \in \mathcal{Z} \right\},$$

where $\boldsymbol{X}(z) = \boldsymbol{a}_{1:n} + \boldsymbol{b}_{1:n}z$, and $\boldsymbol{c}_{1:n}$ represents a vector composed of the first $n$ elements of the vector $\boldsymbol{c}$. The vectors $\boldsymbol{a}, \boldsymbol{b} \in \mathbb{R}^{2n}$ are defined as

$$\boldsymbol{a} = \mathcal{Q}_{\boldsymbol{X}_{\text{obs}}}, \; \boldsymbol{b} = \frac{\tilde{\Sigma}\boldsymbol{\nu}_{\mathcal{M}_{\boldsymbol{X}_{\text{obs}}}}}{\boldsymbol{\nu}_{\mathcal{M}_{\boldsymbol{X}_{\text{obs}}}}^{\top}\tilde{\Sigma}\boldsymbol{\nu}_{\mathcal{M}_{\boldsymbol{X}_{\text{obs}}}}},$$

and the region $\mathcal{Z}$ is defined as

$$\mathcal{Z} = \left\{ z \in \mathbb{R} \mid \mathcal{M}_{\boldsymbol{X}(z)} = \mathcal{M}_{\boldsymbol{X}_{\text{obs}}} \right\}. \qquad (8)$$

Let us consider a random variable $Z \in \mathbb{R}$ and its observation $z_{\text{obs}} \in \mathbb{R}$ so that they satisfy $\boldsymbol{X} = \boldsymbol{a}_{1:n} + \boldsymbol{b}_{1:n}Z$ and $\boldsymbol{X}_{\text{obs}} = \boldsymbol{a}_{1:n} + \boldsymbol{b}_{1:n}z_{\text{obs}}$. Then, the selective $p$-value in (7) is re-written as

$$p_{\text{selective}} = \mathbb{P}_{\text{H}_0}\left(|Z| > |z_{\text{obs}}| \mid Z \in \mathcal{Z}\right). \qquad (9)$$

Under the null hypothesis $\text{H}_0$, the distribution of the unconditional variable $Z$ is $\mathcal{N}(0, \boldsymbol{\nu}_{\mathcal{M}_{\boldsymbol{X}}}^{\top}\tilde{\Sigma}\boldsymbol{\nu}_{\mathcal{M}_{\boldsymbol{X}}})$. Consequently, given $Z \in \mathcal{Z}$, the conditional random variable $Z$ adheres to a truncated Gaussian distribution. Once the truncated region $\mathcal{Z}$ is identified, computing the selective $p$-value in (9) becomes straightforward. Therefore, the remaining task is the identification of $\mathcal{Z}$.

## 4.2 OVER-CONDITIONING

To compute the truncated region $\mathcal{Z}$, we employ a divide and conquer approach. It is difficult to directly identify the truncated region $\mathcal{Z}$ due to the complexity of the computational algorithm of the diffusion model. The basic idea of this approach is to decompose the data space $\mathcal{X}$ into a set of polyhedra by considering additional conditioning, which we refer to as *over-conditioning (OC)* (Duy & Takeuchi, 2022). It is easy to understand that a polyhedron in the $n$-dimensional data space $\mathcal{X}$ corresponds to an interval in the one-dimensional space $\mathcal{Z}$. Therefore, we can sequentially examine intervals in the one-dimensional space and check whether the same hypothesis (anomalous region) as the observed one is selected. In this study, we show that the filtered reconstruction error $E(\boldsymbol{X})$ can be expressed as a piecewise-linear function of $\boldsymbol{X}$. By exploiting this, we identify a over-conditioned interval $\mathcal{Z}^{\text{oc}} \subset \mathcal{Z}$.

**Identification of $\mathcal{Z}^{\text{oc}}$.** Let us write a polyhedron $\mathcal{P}$ composed of piecewise-linear functions as

$$\mathcal{P}_k = \{\boldsymbol{\Delta}_k \boldsymbol{X} \leq \boldsymbol{\delta}_k\}, k \in [K],$$

where $\boldsymbol{\Delta}_k$ and $\boldsymbol{\delta}_k$ for $k \in [K]$ are the coefficient matrix and the constant vector with appropriate dimensions of the $k$-th piecewise-linear function, respectively. Then, a piecewise-linear function $\mathcal{A}(\boldsymbol{X})$ is written in the following form:

$$\mathcal{A}(\boldsymbol{X}) = \begin{cases} \boldsymbol{\Psi}_1 \boldsymbol{X} + \boldsymbol{\psi}_1 & \text{if } \boldsymbol{X} \in \mathcal{P}_1, \\ \boldsymbol{\Psi}_2 \boldsymbol{X} + \boldsymbol{\psi}_2 & \text{if } \boldsymbol{X} \in \mathcal{P}_2, \\ \quad \vdots \\ \boldsymbol{\Psi}_K \boldsymbol{X} + \boldsymbol{\psi}_K & \text{if } \boldsymbol{X} \in \mathcal{P}_K, \end{cases}$$

where $\boldsymbol{\Psi}_k$ and $\boldsymbol{\psi}_k$ for $k \in [K]$ are the coefficient matrix and the constant vector with appropriate dimensions for the $k$-th polyhedron, respectively. Using the notation in (4.1), since the input image $\boldsymbol{X}(z)$ is restricted on a one-dimensional line, each component of the output of $\mathcal{A}$ is written as

$$
\mathcal{A}_i(\boldsymbol{X}(z)) = \begin{cases} \kappa_1^{\mathcal{A}_i} z + \rho_1^{\mathcal{A}_i} & \text{if } z \in [L_1^{\mathcal{A}_i}, U_1^{\mathcal{A}_i}], \\ \kappa_2^{\mathcal{A}_i} z + \rho_2^{\mathcal{A}_i} & \text{if } z \in [L_2^{\mathcal{A}_i}, U_2^{\mathcal{A}_i}], \\ \quad \vdots \\ \kappa_{K(\mathcal{A}_i)}^{\mathcal{A}_i} z + \rho_{K(\mathcal{A}_i)}^{\mathcal{A}_i} & \text{if } z \in [L_{K(\mathcal{A}_i)}^{\mathcal{A}_i}, U_{K(\mathcal{A}_i)}^{\mathcal{A}_i}], \end{cases}
$$

where $K(\mathcal{A}_i)$ is the number of linear pieces of the piecewise-linear function, and $\kappa_k^{\mathcal{A}_i} \in \mathbb{R}$ and $\rho_k^{\mathcal{A}_i} \in \mathbb{R}$ for $k \in [K(\mathcal{A}_i)]$ are the coefficient and the constant of the $k$-th polyhedron, respectively. For each $i \in [n]$, there exists $k \in [K(\mathcal{A}_i)]$ such that $z \in [L_k^{\mathcal{A}_i}, U_k^{\mathcal{A}_i}]$, then the inequality $\mathcal{A}_i(\boldsymbol{X}(z)) \geq \lambda$, can be solved as

$$
[L_z^i, U_z^i] := \begin{cases} \left[ \max \left( L_k^{\mathcal{A}_i}, \left( (\lambda - \rho_k^{\mathcal{A}_i})/\kappa_k^{\mathcal{A}_i} \right) \right), U_k^{\mathcal{A}_i} \right] & \text{if } \kappa_k^{\mathcal{A}_i} > 0, \\ \left[ L_k^{\mathcal{A}_i}, \min \left( U_k^{\mathcal{A}_i}, \left( (\lambda - \rho_k^{\mathcal{A}_i})/\kappa_k^{\mathcal{A}_i} \right) \right) \right] & \text{if } \kappa_k^{\mathcal{A}_i} < 0. \end{cases}
$$

We denote the over-conditioned interval as

$$
\mathcal{Z}^{\mathrm{oc}}(\boldsymbol{a} + \boldsymbol{b}z) = \bigcap_{i \in [n]} \left[ L_z^i, U_z^i \right]. \tag{10}
$$

**Piecewise linearity of diffusion models.** We now show that the diffusion model mapping $\mathcal{D}$ and then filtered reconstruction error $E$ can be expressed as a piecewise-linear function of $\boldsymbol{X}$. To show this, we see that both the forward process and reverse process of the diffusion model are piecewise-linear functions as long as we employ a class of U-Net described below. It is easy to see the piecewise-linearity of the forward process as long as we fix the random seed for $\epsilon_t$. To make the reverse process a piecewise-linear function, we employ a U-Net architecture composed of piecewise-linear components such as ReLU activation function and average pooling. Then, $\epsilon_\theta^{(t)}(\mathbf{x}_t)$ is represented as a piecewise-linear function of $\mathbf{x}_t$. Moreover, since $f_\theta^{(t)}(\mathbf{x}_t)$ in (3) is a composite function of $\epsilon_\theta^{(t)}(\mathbf{x}_t)$, it is also a piecewise-linear function. By combining them together, we see that $\mathbf{x}_{t-1}$ is written as a piecewise-linear function of $\mathbf{x}_t$. Therefore, the entire reconstruction process is a piecewise-linear function since it just repeats the above operation multiple times (see Algorithm 1). As a result, the entire mapping $\mathcal{D}(\boldsymbol{X})$ of the diffusion model is a piecewise-linear function of the input image $\boldsymbol{X}$. Moreover, since the averaging filter $\mathcal{F}$ and the absolute operation are also piecewise-linear functions, $|\mathcal{F}(\boldsymbol{X} - \mathcal{D}(\boldsymbol{X}))|(= E(\boldsymbol{X}))$ is piecewise-linear. By exploiting this piecewise-linearity, the interval $\mathcal{Z}^{\mathrm{oc}}$ can be computed.

### 4.3 IDENTIFICATION OF $\mathcal{Z}$ BY PARAMETRIC PROGRAMMING

Over-conditioning causes a reduction in power due to excessive conditioning. A technique called Parametric Programming is utilized to explore all intervals along the one-dimensional line, resulting in (8). The truncated region $\mathcal{Z}$ can be represented using $\mathcal{Z}^{\mathrm{oc}}$ as

$$
\mathcal{Z} = \bigcup_{z \in \mathbb{R} | \mathcal{M}_{\boldsymbol{X}(z)} = \mathcal{M}_{\boldsymbol{x}_{\mathrm{obs}}}} \mathcal{Z}^{\mathrm{oc}}(\boldsymbol{a} + \boldsymbol{b}z).
$$

The number of $\mathcal{Z}^{\mathrm{oc}}$ is obviously finite due to the finiteness of the number of polyhedra, but for practical purposes it grows exponentially, making it difficult to identify all of them. In many other SI studies, it is known that a search from $z_{\min} = (-10\sigma - |z_{\mathrm{obs}}|)$ to $z_{\max}(= 10\sigma + |z_{\mathrm{obs}}|)$ is sufficient for practical use, where $\sigma$ is the standard deviation of the test statistic $T(\boldsymbol{X}, \boldsymbol{X}^{\mathrm{ref}})$. An algorithm for calculating the selective $p$-value via Parametric Programming is summarized in Algorithm 2.

---

**Algorithm 2** Selective $p$-value Computation by Parametric Programming

---

**Require:** $\boldsymbol{X}_{\mathrm{obs}}, \boldsymbol{X}_{\mathrm{obs}}^{\mathrm{ref}}, z_{\min}, z_{\max}$ and $z_{\mathrm{obs}} := T(\boldsymbol{X}_{\mathrm{obs}}, \boldsymbol{X}_{\mathrm{obs}}^{\mathrm{ref}})$
 1: $\mathcal{Z} \leftarrow \emptyset$
 2: Obtain $\mathcal{M}_{\boldsymbol{X}_{\mathrm{obs}}}$ by (5)
 3: Compute $\boldsymbol{a}, \boldsymbol{b}$ by (8)
 4: $z \leftarrow z_{\min}$
 5: **while** $z < z_{\max}$ **do**
 6:     Compute $\mathcal{Z}^{\mathrm{oc}}(\boldsymbol{a} + \boldsymbol{b}z)$ and $\mathcal{M}_{\boldsymbol{X}(z)}$ by (10) for $z$
 7:     **if** $\mathcal{M}_{\boldsymbol{X}(z)} = \mathcal{M}_{\boldsymbol{X}_{\mathrm{obs}}}$ **then**
 8:         $\mathcal{Z} \leftarrow \mathcal{Z} \cup \mathcal{Z}^{\mathrm{oc}}(\boldsymbol{a} + \boldsymbol{b}z)$
 9:     **end if**
10:     $z \leftarrow \max \mathcal{Z}^{\mathrm{oc}}(\boldsymbol{a} + \boldsymbol{b}z) + \gamma$, where $\gamma$ is small positive number.
11: **end while**
12: $p_{\mathrm{selective}} = \mathbb{P}_{\mathrm{H}_0}\left(|Z| > |z_{\mathrm{obs}}| \mid Z \in \mathcal{Z}\right)$
**Ensure:** $p_{\mathrm{selective}}$

---

## 5 EXPERIMENTS

We compared our proposed methods (DMAD-test, DMAD-test-oc) with the other methods: naive method (naive), bonferroni correction (bonferroni), and permutation test (permutation) on type I error rate and power. The details of the methods for comparison are described in Appendix B. The architecture of the diffusion model used across all experiment settings is detailed in Appendix C. The computation time analysis is presented in Appendix E. We executed the experiment on AMD EPYC 9474F processor, 48-core 3.6GHz CPU and 768GB memory.

### 5.1 NUMERICAL EXPERIMENTS

**Experimental setup.** Experiments on the type I error rate and power were conducted with two types of covariance matrices: independent $\Sigma = I_n \in \mathbb{R}^{n \times n}$ and correlation $\Sigma = (0.5^{|i-j|})_{ij} \in \mathbb{R}^{n \times n}$. In the type I error rate experiments, we used only normal images. The synthetic dataset for normal images is generated to follow $\boldsymbol{X} = (X_1, X_2, \ldots, X_n)^\top \sim \mathcal{N}(\boldsymbol{0}, \Sigma)$. We made 1000 normal images for $n \in \{64, 256, 1024, 4096\}$. In the power experiments, we used only abnormal images. We generated 1000 abnormal images $\boldsymbol{X} = (X_1, X_2, \ldots, X_n)^\top \sim \mathcal{N}(\boldsymbol{\mu}, \Sigma)$. The mean vector $\boldsymbol{\mu}$ is defined as $\mu_i = \Delta$ for all $i \in \mathcal{S}$, and $\mu_i = 0$ for all $i \in [n] \backslash \mathcal{S}$, where $\mathcal{S} \subset [n]$ is the anomalous region with its position randomly chosen. The image size of the abnormal images was set to 4096, with signals $\Delta \in \{1, 2, 3, 4\}$. In all experiments, we made the synthetic dataset for 1000 reference images to follow $\boldsymbol{X}^{\mathrm{ref}} = (X_1^{\mathrm{ref}}, X_2^{\mathrm{ref}}, \ldots, X_n^{\mathrm{ref}})^\top \sim \mathcal{N}(\boldsymbol{0}, \Sigma)$. The threshold was set to $\lambda = 0.8$, and the kernel size of the averaging filter was set to 3. All experiments were conducted under the significance level $\alpha = 0.05$. The diffusion models were trained on the normal images from the synthetic dataset. The diffusion models were trained with $T = 1000$ and the initial time step of the reverse process was set to $T' = 460$, and the reconstruction was conducted 5 step samplings. The noise schedule $\beta_1, \beta_2, \ldots, \beta_T$ was set to linear. In all experiment, we aim to generate new images through probabilistic sampling, $\eta$ was set to 1. In addition, we conducted robustness experiments against non-Gaussian noise. The details of the robustness experiments are described in Appendix D.

**Results.** Figures 2a and 2b show the comparison results of type I error rates. The proposed methods DMAD-test and DMAD-test-oc can control the type I error rate at the significance level $\alpha$, and bonferroni can control the type I error rate below the significance level $\alpha$. In contrast, naive and permutation cannot control the type I error rate. Figures 2c and 2d show the comparison results of powers. Since naive and permutation cannot control the type I error rate, their powers are not considered. Among the methods that can control the type I error rate, the proposed method has the highest power. DMAD-test-oc is over-conditioned and bonferroni is conservative because there are many hypotheses, so they have low power. Figure 3 shows the results of the robustness experiments. DMAD-test maintains good performance on the type I error rate for all the considered distribution families.

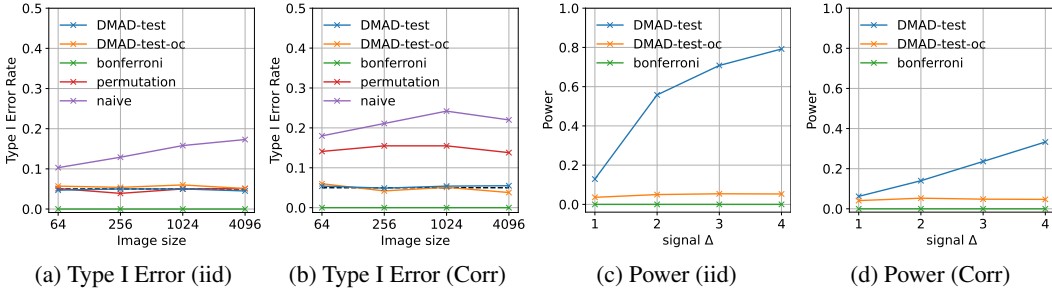

(a) Type I Error (iid)    (b) Type I Error (Corr)    (c) Power (iid)    (d) Power (Corr)

Figure 2: Comparison of Type I Error Rate and Power. Figures (a) and (b) show type I error rates, while (c) and (d) show power under independence (iid) and correlation (Corr) noise settings. Only the proposed method and the bonferroni correction successfully control type I error rates. The `DMAD-test` has the highest power among the methods that can control the type I error rate.

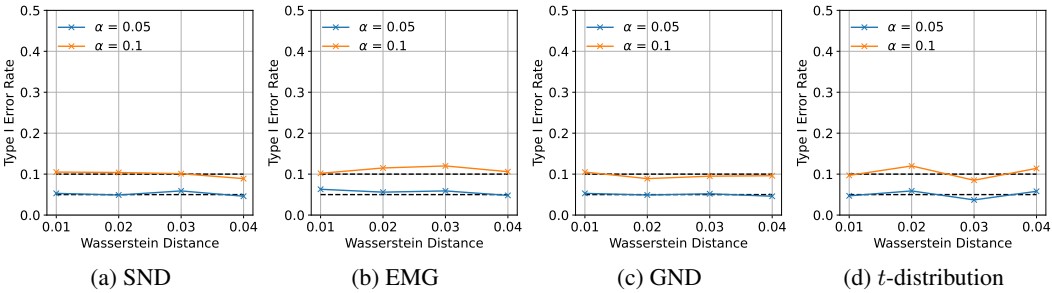

(a) SND    (b) EMG    (c) GND    (d) $t$-distribution

Figure 3: Type I Error Rate of the `DMAD-test` for Non-Gaussian Distribution Families. The `DMAD-test` exhibits robust performance.

## 5.2 REAL DATA EXPERIMENTS

We conducted experiments using T2-FLAIR MRI brain scans from the Brain Tumor Segmentation (BraTS) 2023 dataset (Karargyris et al., 2023; LaBella et al., 2023), which comprises 934 non-skull-stripped 3D scans with dimensions of $240 \times 240 \times 155$. From these scans, we extracted 2D $240 \times 240$ axial slices at axis 95, resized them to $64 \times 64$ pixels, and categorized them based on the ground truth annotations into 532 normal images (without tumors) and 402 abnormal images (with tumor regions). For each scan, we estimated the mean and variance from pixel values excluding both the non-brain regions and tumor regions identified in the ground truth, followed by standardization. We randomly selected 312 normal images for model training. The model was trained with $T = 1000$ and the initial time step of the reverse process was set at $T' = 300$, with reconstruction performed through 5 step samplings. We set the threshold $\lambda = 0.85$ and the kernel size of the averaging filter to 3. Note that, when testing images, the non-brain regions are not treated as anomalous regions $\mathcal{M}_{\boldsymbol{X}}$. The results of the `DMAD-test` and `naive` are shown in the Figure 4. The naive $p$-values are low for both normal and abnormal images, while the selective $p$-values are high for normal images and low for abnormal images. This result indicates that the `DMAD-test` detected anomalous regions as statistically significant while avoiding misidentification of normal image as anomaly.

## 6 CONCLUSIONS

In this study, we proposed a novel statistical test for anomalous regions in medical images detected by using a diffusion model. With the proposed DMAD-test, the false detection rate can be controlled with the significance level because statistical inference is conducted conditional on the fact that the anomalous regions are identified by using a diffusion model. We believe this study marks a step toward bridging the gap between generative AI and rigorous statistical inference in medical imaging analysis.

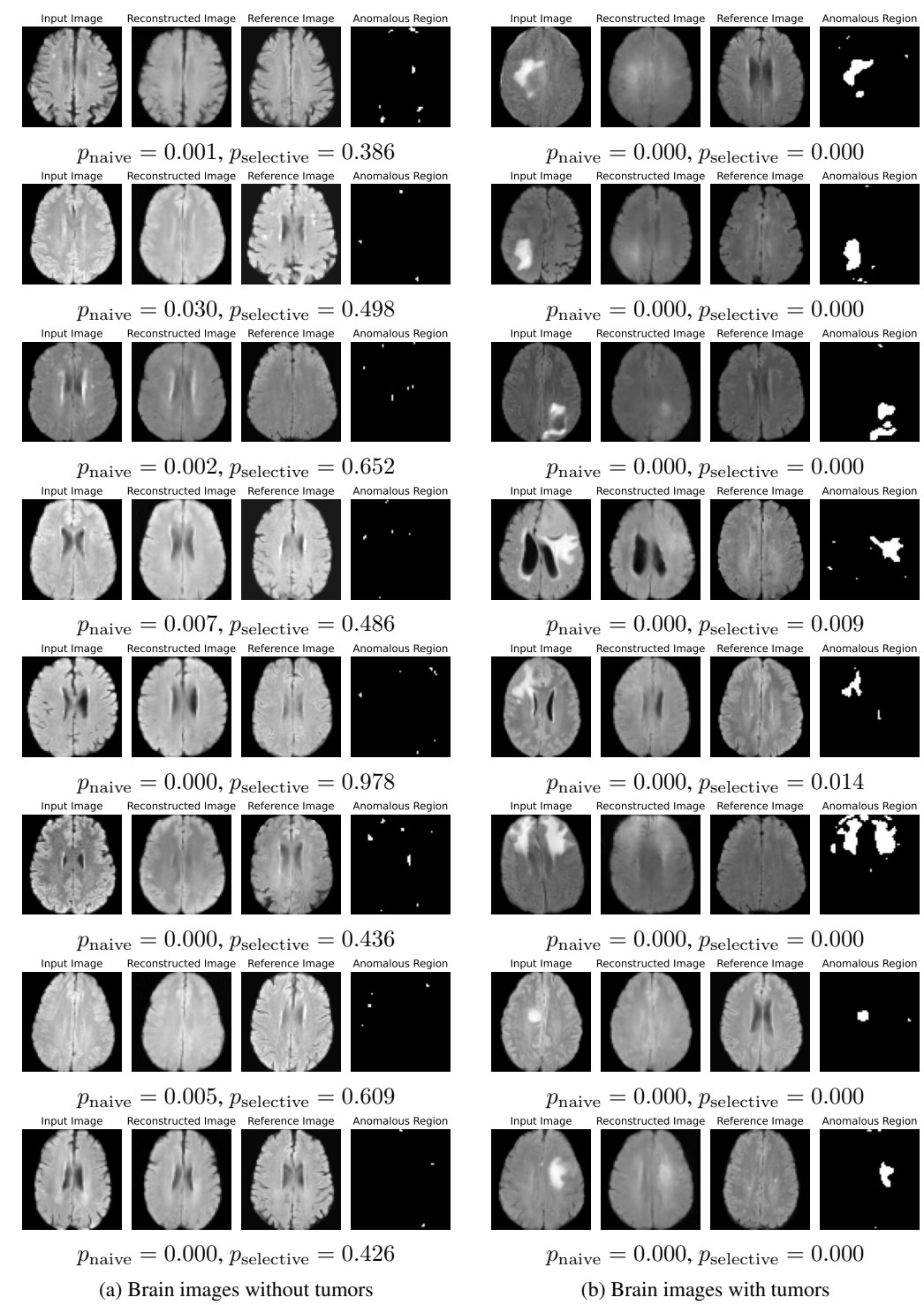

(a) Brain images without tumors       (b) Brain images with tumors

Figure 4: An example of the results for applying the proposed DMAD-test and the naive test (an invalid test ignoring that the anomalous region was identified by the diffusion model) to brain images. The left column represents the results for normal brain images without tumors, while the right column represents the results for abnormal brain images with tumors. The $p_{\mathrm{selective}}$ calculated by the proposed DMAD-test is high for normal images (True Negative) and low for abnormal images (True Positive), indicating that the results are desirable. On the other hand, the $p_{\mathrm{naive}}$ obtained by the naive test is low not only for abnormal images but also for normal images (False Positive), indicating the invalidness of the naive test.

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

## A  ACCELERATED REVERSE PROCESSES

Methods for accelerating the reverse process have been proposed in DDPM, DDIM (Song et al., 2022). When taking a strictly increasing subsequence $\tau$ from $\{1, \cdots, T\}$, it is possible to skip the sampling trajectory from $\mathbf{x}_{\tau_i}$ to $\mathbf{x}_{\tau_{i-1}}$. In this case, equations (2) and (4) can be rewritten as

$$\mathbf{x}_{\tau_{i-1}} = \sqrt{\alpha_{\tau_{i-1}}}\left(\frac{\mathbf{x}_{\tau_i} - \sqrt{1-\alpha_{\tau_i}} \cdot \epsilon^{(\tau_i)}(\mathbf{x}_{\tau_i})}{\sqrt{\alpha_{\tau_i}}}\right) + \sqrt{1-\alpha_{\tau_{i-1}} - \sigma_{\tau_i}^2} \cdot \epsilon_\theta^{(\tau_i)}(\mathbf{x}_{\tau_i}) + \sigma_{\tau_i}\epsilon_{\tau_i},$$

where

$$\sigma_{\tau_i} = \eta\sqrt{(1-\alpha_{\tau_{i-1}})/(1-\alpha_{\tau_i})}\sqrt{1-\alpha_{\tau_i}/\alpha_{\tau_{i-1}}}.$$

Therefore, piecewise-linearity is preserved, making the proposed method DMAD-test applicable.

## B COMPARISON METHODS FOR NUMERICAL EXPERIMENTS

We compared our proposed method with the following methods:

- `DMAD-test`: The proposed method uses the parametric programming.

- `DMAD-test-oc`: The proposed method uses over-conditioning.

- `naive`: The naive method. This method uses a conventional $z$-test without any conditioning. The naive $p$-value is calculated as

$$p_{\text{naive}} = \mathbb{P}_{\mathbb{H}_0} \left( |Z| > |z_{\text{obs}}| \right).$$

- `bonferroni`: To control the type I error rate, this method applies the bonferroni correction. Given that the total number of anomaly regions is $2^n$, the $p$-value is calculated as ,

$$p_{\text{bonferroni}} = \min(1, 2^n \cdot p_{\text{naive}}).$$

- `permutation`: This method uses a permutation test with the steps outlined below:

  - Calculate the observed test statistic $z_{\text{obs}}$ by applying the observed image $\boldsymbol{X}_{\text{obs}}$ to the diffusion model.
  - For each $i = 1, \ldots, B$, compute the test statistic $z^{(i)}$ by applying the permuted image $\boldsymbol{X}^{(i)}$ to the diffusion model, where $B$ represents the total number of permutations, set to 1,000 in our experiments.
  - The permutation $p$-value is then determined as

$$p_{\text{permutation}} = \frac{1}{B} \sum_{b \in [B]} \mathbf{1}\{|z^{(b)}| > |z_{\text{obs}}|\},$$

  where $\mathbf{1}\{\cdot\}$ denotes the indicator function.

This rephrasing aims to maintain the original meaning while enhancing readability and comprehension.

## C ARCHITECTURE OF THE U-NET

Figure 5 shows the architecture of the U-Net used in our experiments. The U-Net has three skip connections, and the Encoder and Decoder blocks. For image sizes $n \in \{64, 256, 1024, 4096\}$, the corresponding spatial dimensions of images are $(1, d, d)$ where $d \in \{8, 16, 32, 64\}$.

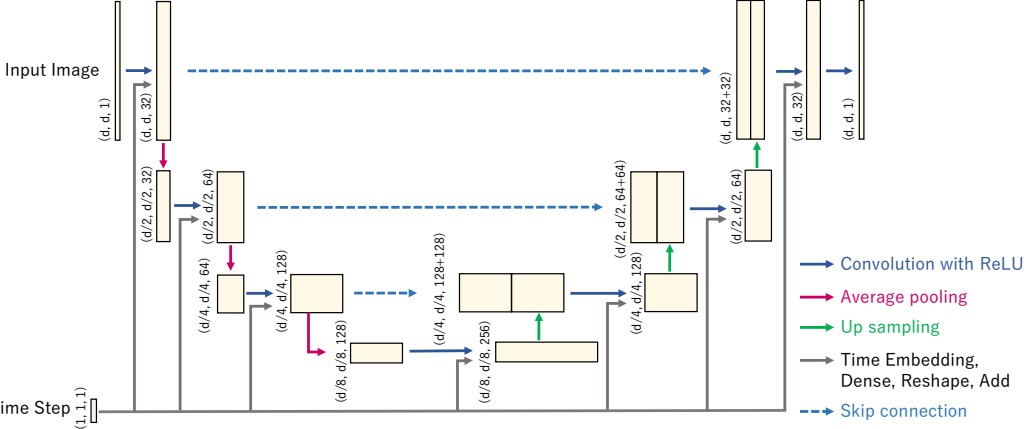

Figure 5: The architecture of the U-Net

## D    ROBUSTNESS OF THE PROPOSED METHOD

To evaluate the robustness of our proposed method's performance, we used various non-Gaussian distribution families with different levels of deviation from the standard normal distribution $\mathcal{N}(0, 1)$. We considered the following non-Gaussian distributions with a 1-Wasserstein distance $d \in \{0.01, 0.02, 0.03, 0.04\}$ from $\mathcal{N}(0, 1)$:

- Skew normal distribution family (SND).

- Exponentially modified gaussian distribution family (EMG).

- Generalized normal distribution family (GND) with a shape parameter $\beta$. This distribution family can be steeper than the normal distribution (i.e., $\beta < 2$).

- Student's $t$-distribution family ($t$-distribution).

Note that these distributions are standardized in the experiments. Figure 6 shows the probability density functions for distributions from each family, such that the $d$ is set to $0.04$. We run 1000 trials for each distribution family and each 1-Wasserstein distance to calculate the type I error rate. The significance levels $\alpha$ were set to $0.05$ and $0.10$, and the image size was set to $256$.

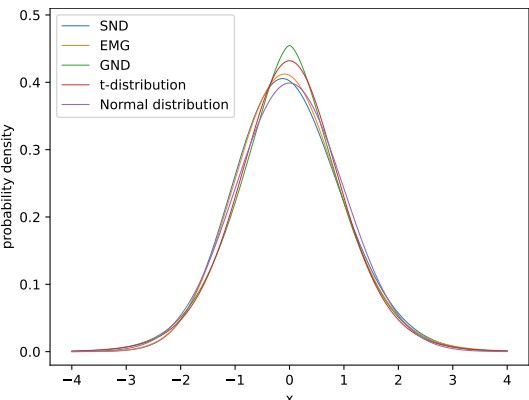

Figure 6: Non-Gaussian distributions with $d = 0.04$

## E    COMPUTATION TIME ANALYSIS

We conducted a comprehensive evaluation of the computation times for the proposed method `DMAD-test` using an AMD EPYC 9474F processor (48-core, 3.6GHz). Figure 7 shows the computation time when changing the image size for the synthetic data. These experiments were conducted under the same settings as the type I error rate experiments described in §5.1. To optimize performance, we applied an acceleration technique that enables early termination once $p$-values reach sufficient precision. The detail of this technique is described in Shiraishi et al. (2024a). Theoretically, while the number of intervals on a one-dimensional line should scale exponentially with image size, our empirical results demonstrate substantially better practical performance. Table 1 shows the computation times for the brain image dataset described in §5.2, where the times were averaged over 100 images each of brains with and without tumor. We performed interval calculations for the $p$-value in parallel using 48 cores in this experiment. The computation time was 1100 seconds per image without tumors and 4220 seconds per image with tumors, demonstrating the method's feasibility for clinical applications.

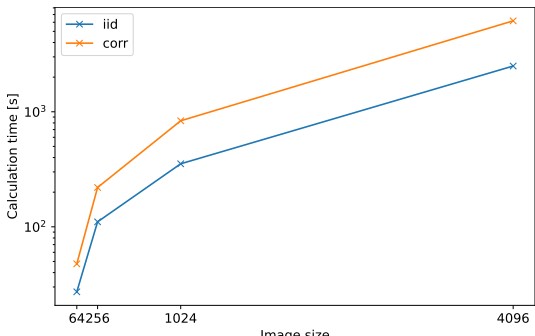

Figure 7: Computation time when changing the image size for the synthetic data. Results are shown for both synthetic data with independent (iid) and correlation (corr) noise.

Table 1: Computation time for brain images using parallel processing across 48 cores.

| Image | Time (s) |
|---|---|
| Brain image without tumors | 1100 |
| Brain image with tumors | 4220 |