# OpenReview forum: "Statistical Test on Diffusion Model-based Anomaly Detection by Selective Inference"
_ICLR.cc/2025/Conference — ICLR 2025 Conference Withdrawn Submission_

### Official Review · Reviewer_NqFB · 2024-10-27

**Soundness:** 3
**Presentation:** 3
**Contribution:** 3
**Rating:** 5
**Confidence:** 3

**Summary:**

This paper introduces a statistical framework for anomaly detection in medical images using diffusion models, emphasizing reliable decision-making through selective inference. The authors propose a Diffusion Model-based Anomalous Region Detection test, which quantifies the statistical reliability of detected anomalies by computing selective p-values. This approach allows for controlled error rates, supporting clinical applications where rigorous reliability is essential. They implement selective inference with a piecewise-linear reconstruction error function, leveraging polyhedral geometry, and validate their method through numerical experiments and brain imaging tests. The work aims to bridge the gap between generative AI and statistical rigor in medical diagnostics.

**Strengths:**

This approach leverages the unique combination of polyhedral and piecewise-linear functions in constructing a selective p-value, which allows the method to yield statistically sound inferences in an area where conventional anomaly detection techniques might lack formal reliability measures. The use of selective inference in the context of diffusion-based image generation is particularly novel, as the approach traditionally applies to feature selection in simpler models.

The authors provide a thorough theoretical foundation for their test. Their experiments demonstrate both the effectiveness and robustness of the method, detailing performance under various types of noise distributions and comparing it against other statistical tests like Bonferroni correction and permutation testing. Additionally, the stepwise construction and conditioning processes for selective inference are well-documented, highlighting the method’s robustness and adaptability in different testing scenarios.

The paper is well-structured, with a logical progression from the theoretical underpinnings to the experimental validation. The core concepts, including the their test, selective p-value, and the role of selective inference, are clearly defined. Figures illustrating the architecture and reconstruction process enhance reader comprehension. The piecewise-linear and polyhedral properties are described in a way that allows technically proficient readers to understand the model’s behavior and interpretability.

This work has some potential for impact. By providing statistical reliability measures via selective inference, the authors address a key limitation in deploying AI for clinical diagnostics: the need for transparent and error-controlled anomaly detection. The methodology’s ability to maintain controlled false-positive rates aligns with clinical requirements, supporting better trust and broader adoption of generative AI in medical applications. This is a meaningful contribution.

**Weaknesses:**

The statement "the diffusion model detects anomalies" (p.4) is somewhat misleading, as the detection is more accurately based on comparing input and reconstructed images to highlight discrepancies. This phrasing implies the model actively detects anomalies, while in practice, it passively reconstructs images, with anomalies inferred from reconstruction errors. A more accurate description could clarify the diffusion model's role and prevent misinterpretation of the model's capabilities.

Many plots, notably Figures 2, 3, and 5, lack essential elements like axis labels and legends, limiting their interpretability. For complex analyses like these, well-labeled figures are critical for readers to follow and validate the findings.

Given the complexity of the statistical testing framework, a visual illustration of the DMAD-test would be beneficial. Many statistical papers can be challenging to interpret without sufficient diagrams, and a schematic representation of the test could improve comprehension, particularly for readers unfamiliar with selective inference. This could be placed early on to guide readers through the methodology.

While computational resources are mentioned, there is no discussion of computational time, which is critical for assessing the method’s feasibility in practical applications. Adding specific timing metrics, ideally with a breakdown by test stage, would help readers gauge whether this approach could realistically be applied to clinical workflows.

The experiments focus on brain images with tumors, limiting the generalizability of the findings. Testing on a broader dataset, such as DeepLesion (https://nihcc.app.box.com/v/DeepLesion), or including non-tumor regions within volumes could provide a more comprehensive evaluation. Additionally, validating on datasets with a mix of normal and abnormal slices would assess the model's robustness and applicability to varied clinical scenarios.

The supplementary material mentions code, but it is unclear if this is executable code or pseudocode. Given the complexity of the statistical framework, accessible and runnable code would support reproducibility and encourage other researchers to build on this work.

While the theoretical equations appear solid, some expressions, especially those in the SI framework, are intricate. Although no clear mistakes are present, a supplementary derivation or walkthrough of the core equations would support readers who may struggle with verifying them. This could be presented as an appendix to guide readers through critical steps and foster better understanding.

**Questions:**

- Could you clarify the reasoning behind describing the diffusion model as “detecting” anomalies?
- Could you provide details on the computational time required for the DMAD-test on a standard dataset?
- Could you clarify whether the supplementary code provided is fully executable, and, if so, whether it includes detailed instructions for setup and testing?
- Have you considered testing on broader datasets, such as DeepLesion or other modalities?
- Given the computational complexity and reliance on high-level statistical inference, do you foresee any specific challenges in deploying this method in real-time clinical settings?

---

> ### Author Response · Authors · 2024-11-25
>
> We appreciate the reviewer's careful review and thoughtful comments. We provide answers to the specific concerns.
>
> > The statement "the diffusion model detects anomalies" (p.4) is somewhat misleading, as the detection is more accurately based on comparing input and reconstructed images to highlight discrepancies. This phrasing implies the model actively detects anomalies, while in practice, it passively reconstructs images, with anomalies inferred from reconstruction errors. A more accurate description could clarify the diffusion model's role and prevent misinterpretation of the model's capabilities.
>
> We appreciate the reviewer's thoughtful comment. We revised potentially misleading expressions in the revised version.
>
> > Many plots, notably Figures 2, 3, and 5, lack essential elements like axis labels and legends, limiting their interpretability. For complex analyses like these, well-labeled figures are critical for readers to follow and validate the findings.
>
> We are afraid that the explanation of our figures was insufficient and failed to effectively convey our intention. We have added explanations to the parts of the figures that were deemed difficult to understand. Specifically, for Figure 4's power comparison, we deliberately excluded "naive" and "permutation" methods because, as demonstrated in Figure 3's type I error rate comparison, these methods cannot control the type I error rate. It is not appropriate to consider the power of methods that fail to maintain type I error control. Additionally, in Figure 5's robustness experiments, we specifically focused on verifying the robustness of type I error rate control, which is a critical aspect of statistical validity. In revised version, we emphasize these points more clearly.
>
> > Given the complexity of the statistical testing framework, a visual illustration of the DMAD-test would be beneficial. Many statistical papers can be challenging to interpret without sufficient diagrams, and a schematic representation of the test could improve comprehension, particularly for readers unfamiliar with selective inference. This could be placed early on to guide readers through the methodology.
>
> We appreciate your suggestion and enhance Figure 1 to provide a more detailed overview of the DMAD-test methodology in the revised version.
>
> > While computational resources are mentioned, there is no discussion of computational time, which is critical for assessing the method’s feasibility in practical applications. Adding specific timing metrics, ideally with a breakdown by test stage, would help readers gauge whether this approach could realistically be applied to clinical workflows.
>
> We will add a comprehensive analysis of computational time in the Appendix, including a breakdown of computational times for each stage. While computational time is a consideration for real-time clinical deployment, our method can be optimized through parallel processing.
>
> > The experiments focus on brain images with tumors, limiting the generalizability of the findings. Testing on a broader dataset, such as DeepLesion (https://nihcc.app.box.com/v/DeepLesion), or including non-tumor regions within volumes could provide a more comprehensive evaluation. Additionally, validating on datasets with a mix of normal and abnormal slices would assess the model's robustness and applicability to varied clinical scenarios.
>
> The brain image experiments were intended as a demonstration of our method's applicability rather than a comprehensive comparison with existing approaches. It should be noted that our primary contribution lies NOT in proposing a new anomaly detection method, but in accurately evaluating the statistical significance of existing anomaly detection methods. Methods that can theoretically ensure the correct evaluation of the statistical significance of detected anomaly regions are currently limited to overly conservative multiple testing corrections (e.g., Bonferroni Method). In this paper, we demonstrate the quantitative effectiveness of the proposed method in synthetic data experiments, where the true signal and noise are provided as oracles, by showing that it can control type I error (false positive detection probability) at a specified significance level and achieves significantly higher detection power than the Bonferroni correction.

---

> > ### Author Response · Authors · 2024-11-25
> >
> > > The supplementary material mentions code, but it is unclear if this is executable code or pseudocode. Given the complexity of the statistical framework, accessible and runnable code would support reproducibility and encourage other researchers to build on this work.
> >
> > We have provided fully executable code in the supplementary material. While basic setup instructions are included in the README.md, we enhance the documentation with more detailed instructions for setup and testing in the revised version.
> >
> > > While the theoretical equations appear solid, some expressions, especially those in the SI framework, are intricate. Although no clear mistakes are present, a supplementary derivation or walkthrough of the core equations would support readers who may struggle with verifying them. This could be presented as an appendix to guide readers through critical steps and foster better understanding.
> >
> > Comprehensively explaining the theory of Selective Inference in this paper is difficult; therefore, we have structured the paper to enhance understanding by outlining the basic concepts and appropriately citing relevant literature. In particular, referring to Lee et al. (2016) provides valuable insight into the theoretical foundation of Selective Inference.
> >
> > > Could you clarify the reasoning behind describing the diffusion model as “detecting” anomalies?
> >
> > The terminology will be revised to more accurately reflect that anomalies are detected through reconstruction error analysis rather than by the diffusion model directly.
> >
> > > Could you provide details on the computational time required for the DMAD-test on a standard dataset?
> >
> > For example, in our current experiments, the DMAD-test requires approximately 40 minutes to compute the p-value for a synthetic normal image (iid) with the largest image size (n = 4096) using a single CPU core. However, it is worth noting that, as our method can be easily parallelized, this computational time can be significantly reduced by leveraging multiple cores.
> >
> > > Could you clarify whether the supplementary code provided is fully executable, and, if so, whether it includes detailed instructions for setup and testing?
> >
> > We have provided fully executable code in the supplementary material. While we currently include basic executable instructions in the README.md file, we will enhance the documentation with more detailed setup and testing procedures in the revised version.
> >
> > > Have you considered testing on broader datasets, such as DeepLesion or other modalities?
> >
> > The brain image experiments were intended as a demonstration of our method's applicability rather than a comprehensive comparison with existing approaches. It should be noted that our primary contribution lies NOT in proposing a new anomaly detection method, but in accurately evaluating the statistical significance of existing anomaly detection methods. Methods that can theoretically ensure the correct evaluation of the statistical significance of detected anomaly regions are currently limited to overly conservative multiple testing corrections (e.g., Bonferroni Method). In this paper, we demonstrate the quantitative effectiveness of the proposed method in synthetic data experiments, where the true signal and noise are provided as oracles, by showing that it can control type I error (false positive detection probability) at a specified significance level and achieves significantly higher detection power than the Bonferroni correction.
> >
> > > Given the computational complexity and reliance on high-level statistical inference, do you foresee any specific challenges in deploying this method in real-time clinical settings?
> >
> > While computational time could present challenges for real-time deployment, we believe this can be addressed through parallel processing optimization. We will include a detailed discussion of these considerations in the revised version and update our supplementary code to demonstrate these optimizations.

---

> > > ### Comment · Reviewer_NqFB · 2024-11-29
> > > **framework is promising but the paper falls short of rigour**
> > >
> > > I apologize for initially misinterpreting the intent of this paper. However, if the primary goal is 'accurately evaluating the statistical significance of existing anomaly detection methods,' then a more comprehensive evaluation of the current state-of-the-art methods in this rapidly evolving field is necessary. Since 2021, significant advancements have been made, yet the paper seems to focus primarily on methods from that time and earlier. I would be particularly interested to see how the proposed approach performs on more recent methods emerging from the CVPR and MICCAI communities.
> > >
> > > While the concept of providing a framework for such evaluations is promising, in its current form, the paper requires substantial revision to meet the rigor needed for thorough testing of existing methods. The depth and scope of the work would likely make it more suitable for a comprehensive journal article, where the authors could provide an extensive overview and evaluation of contemporary methods.
> > >
> > > Under these circumstances, I regretfully must lower my score. However, I wish the authors the best in further developing this work, as I believe it has the potential to become a significant contribution to the field.

---

> > > > ### Author Response · Authors · 2024-12-02
> > > >
> > > > > However, if the primary goal is 'accurately evaluating the statistical significance of existing anomaly detection methods,' then a more comprehensive evaluation of the current state-of-the-art methods in this rapidly evolving field is necessary. Since 2021, significant advancements have been made, yet the paper seems to focus primarily on methods from that time and earlier. I would be particularly interested to see how the proposed approach performs on more recent methods emerging from the CVPR and MICCAI communities. While the concept of providing a framework for such evaluations is promising, in its current form, the paper requires substantial revision to meet the rigor needed for thorough testing of existing methods. The depth and scope of the work would likely make it more suitable for a comprehensive journal article, where the authors could provide an extensive overview and evaluation of contemporary methods.
> > > >
> > > > While we deeply appreciate your response and evaluation in the highest regard, we cannot agree with this comment. We are aware that DDPM has been extended in various directions over the past few years. However, as DDPM remains widely used as the foundation of diffusion models in many practical studies, we believe that it is reasonable and appropriate to use DDPM as an example to demonstrate our statistical testing method. Our contribution in this study lies in providing the theory and the algorithm, and the proposed statistical testing framework can be readily applied even with more advanced diffusion models. We would greatly appreciate it if the reviewer could reconsider the evaluation after reviewing Sections 3 and 4.1 to recognize that our statistical testing framework is not specific to a particular DDPM structure but is generally applicable.

---

### Official Review · Reviewer_gG2v · 2024-11-02

**Soundness:** 2
**Presentation:** 3
**Contribution:** 2
**Rating:** 5
**Confidence:** 4

**Summary:**

This paper proposes a method for detecting and quantify the reliability of anomalous regions in images. In particular, the authors focus on images generated by diffusion model and use selective inference framework. The authors test the present method using numerical experiments and on a brain image dataset. The output of the proposed model is a p-value that can determine the significance of the detected anomaly.

**Strengths:**

1) The authors address a highly relevant problem by proposing a method for anomaly detection. A method that identifies the significance of anomalies and enable automated anomaly detection in a hospital setting would be extremely useful and important.
2) The authors test their proposed DMAD-test through numerical experiments and on a brain imaging dataset, which provides a good way of evaluating the method's effectiveness in different contexts.

**Weaknesses:**

3) The section on diffusion models feels somewhat out of place, as it mainly covers basic concepts without providing specifics about the model actually used. It may be more effective to reference the foundational paper on diffusion models and focus on describing details relevant to this study, such as was the same architecture of the diffusion model used for the real data and the experimental data? Could the authors provide more information on the model architecture, input image sizes, and preprocessing steps?
4) Expanding on the previous point, the quality of reconstructed images is essential for the accuracy of the resulting p-values. Reporting image quality metrics like MMD (Maximum Mean Discrepancy), SSIM, or MS-SSIM would help assess the reconstruction quality. Specifically, calculating MMD/MS-SSIM between reconstructions and real healthy images would be informative, and it could also be insightful to compare these values between reconstructions of healthy images and those containing anomalies. For reference, Table 1 of https://arxiv.org/pdf/2307.15208 provides a similar evaluation. Including these metrics would significantly strengthen the paper’s conclusions, especially considering that only 329 images were used to train the diffusion model on real data.
5) Additional details on the images used would help readers better understand the study’s setup when using the brain images. Are the images 2D or 3D brain images, and how were they pre-processed before analysis?

**Questions:**

6) The submission used the ICRL 2024 template. Could the authors update the submission to use the ICRL 2025 template?
7) Could the authors elaborate on what is the reference image? How is the reference image different than the reconstructed?
8) Could the authors discuss their choice to focus on diffusion models rather than exploring alternative models like transformers? Why were diffusion models selected over transformers, could the authors compare the advantages and disadvantages of diffusion models versus transformers for this particular anomaly detection task? Could the authors discuss any prior work that has used transformers for similar tasks.

---

> ### Author Response · Authors · 2024-11-25
>
> We appreciate the reviewer's careful review and thoughtful comments. We provide answers to the specific concerns.
>
> > 3. The section on diffusion models feels somewhat out of place, as it mainly covers basic concepts without providing specifics about the model actually used. It may be more effective to reference the foundational paper on diffusion models and focus on describing details relevant to this study, such as was the same architecture of the diffusion model used for the real data and the experimental data? Could the authors provide more information on the model architecture, input image sizes, and preprocessing steps?
>
> We appreciate the reviewer's thoughtful comments. While we explained the basic concepts of diffusion models, we intentionally kept our description architecture-agnostic, since our method can potentially be applied to various diffusion model architectures. We use the standard DDPM's sampling algorithm as described in the paper, with detailed model architecture provided in Appendix C.
>
> > 4. Expanding on the previous point, the quality of reconstructed images is essential for the accuracy of the resulting p-values. Reporting image quality metrics like MMD (Maximum Mean Discrepancy), SSIM, or MS-SSIM would help assess the reconstruction quality. Specifically, calculating MMD/MS-SSIM between reconstructions and real healthy images would be informative, and it could also be insightful to compare these values between reconstructions of healthy images and those containing anomalies. For reference, Table 1 of https://arxiv.org/pdf/2307.15208 provides a similar evaluation. Including these metrics would significantly strengthen the paper’s conclusions, especially considering that only 329 images were used to train the diffusion model on real data.
>
> We appreciate the reviewer's suggestion. While image reconstruction quality is important for clinical applications, our primary contribution is not about improving reconstruction quality but rather providing statistical guarantees for anomaly detection through p-values. Our method's key strength is its ability to control the type I error rate regardless of how the underlying model was trained. This theoretical guarantee of statistical validity holds independent of the specific training process or model architecture used.
>
> > 5. Additional details on the images used would help readers better understand the study’s setup when using the brain images. Are the images 2D or 3D brain images, and how were they pre-processed before analysis?
>
>
>
> We appreciate the reviewer's suggestion. To help readers better understand our experimental setup, we have adopted the Brain Tumor Segmentation (BraTS) 2023 dataset, which is the most widely used standard dataset for 3D MRI brain scans. We extract 2D axial slices from the 3D MRI scans for our experiments and provide detailed preprocessing procedures in the revised version.
>
> > 6. The submission used the ICRL 2024 template. Could the authors update the submission to use the ICRL 2025 template?
>
> We apologize for using the ICLR 2024 template and updated to the 2025 template.
>
> > 7. Could the authors elaborate on what is the reference image? How is the reference image different than the reconstructed?
>
> The reconstructed image is a virtual normal image generated by the diffusion model, representing what the input image would look like if the patient were healthy. By comparing this reconstructed image with the input image, we can identify anomalous regions through reconstruction errors. To quantify the statistical significance of these detected regions, we then compare them with a reference image - an actual normal image obtained from a different healthy subject. This approach, using real normal images as reference, aligns with common medical practice. It should be noted that our contribution does NOT lie in proposing this anomaly detection method BUT in proposing a statistical testing method that can quantify the type I error of the detected anomaly regions.
>
> > 8. Could the authors discuss their choice to focus on diffusion models rather than exploring alternative models like transformers? Why were diffusion models selected over transformers, could the authors compare the advantages and disadvantages of diffusion models versus transformers for this particular anomaly detection task? Could the authors discuss any prior work that has used transformers for similar tasks.
>
> In medical image generation tasks, diffusion models have emerged as one of the state-of-the-art approaches. While transformer-based architectures also exist, extending our statistical testing framework to such architectures remains an open problem for future research.

---

### Official Review · Reviewer_fh2u · 2024-11-04

**Soundness:** 2
**Presentation:** 2
**Contribution:** 2
**Rating:** 5
**Confidence:** 1

**Summary:**

This paper introduces a statistical significance test to reconstruction-based anomaly detection, enhancing the reliability of decision-making with synthetic images generated by AI. In the context of anomaly detection, the proposed DMAD-test method quantifies the statistical significance of anomalous regions using p-values, which helps reduce the false alarm rate. In the experimental section, DMAD-test and its variant, DMAD-test-oc, are evaluated on a synthetic dataset and a brain anomaly detection dataset.

**Strengths:**

1. The paper motivation is well presented. The problem the paper tackles is very important, as generative models may lead to deteriorate performance due to the quality of its output.

2. The introduction of a statistical significance test to refine identified anomaly regions is quite novel, offering a new approach to anomaly localization.

**Weaknesses:**

1. The statistical test relies solely on the mean value of the identified anomalous region, without accounting for structural differences. In medical image anomaly detection, anomalous regions may have mean values similar to those of normal regions; however, the spatial structure of tissues and anatomical patterns often show significant differences. Without considering the structure difference in the statistical test, the obtained results may still suffer from large false detection.

2. It is not clear how to generate the reference image for calculating the statistics.

3. Although the paper includes visualization examples for brain image anomaly detection, it lacks quantitative results. Without a thorough quantitative comparison to prior methods, it is difficult to assess the significance and effectiveness of the proposed approach.

**Questions:**

Please refer to the weakness section for my questions and comments.

---

> ### Author Response · Authors · 2024-11-25
>
> We appreciate the reviewer's careful review and thoughtful comments. We provide answers to the specific concerns.
>
> > 1. The statistical test relies solely on the mean value of the identified anomalous region, without accounting for structural differences. In medical image anomaly detection, anomalous regions may have mean values similar to those of normal regions; however, the spatial structure of tissues and anatomical patterns often show significant differences. Without considering the structure difference in the statistical test, the obtained results may still suffer from large false detection.
>
> We appreciate the reviewer's thoughtful comments regarding structural differences in medical image anomaly detection. In this paper, we focus on the simplest application of using the mean difference. However, our framework can be easily extended to incorporate structural differences by introducing spatial filtering techniques. More specifically, let $H \in \mathbb{R}^{n \times n}$ denote a linear filter operator (e.g., a Laplacian filter). Such filters can help highlight edges and capture structural variations between normal and anomalous regions. In this case, the distribution of the test statistic can be expressed as $\eta_{M_X} H X \sim N(0, (\eta_{M_X} H)^\top \Sigma (\eta_{M_X} H)))$, and it can be easily shown that our proposed method remains valid in this case. We acknowledge that this as an important direction for future work.
>
> > 2. It is not clear how to generate the reference image for calculating the statistics.
>
> As the reviewer pointed out, the role and details of the reference image should have been described more clearly. In our current two-sample test setting, the reference image is randomly selected from normal images that are separate from the test image dataset. Regarding the selection method of the reference images, various other options can be considered. For example, our proposed method can be extended that doctors can select reference images that share similar characteristics with the test image in medical practice. We added a detailed explanation of the reference image selection process in the revised version.
>
> > 3. Although the paper includes visualization examples for brain image anomaly detection, it lacks quantitative results. Without a thorough quantitative comparison to prior methods, it is difficult to assess the significance and effectiveness of the proposed approach.
>
> The brain image experiments were intended as a demonstration of our method's applicability rather than a comprehensive comparison with existing approaches. It should be noted that our primary contribution lies NOT in proposing a new anomaly detection method, but in accurately evaluating the statistical significance of existing anomaly detection methods. Methods that can theoretically ensure the correct evaluation of the statistical significance of detected anomaly regions are currently limited to overly conservative multiple testing corrections (e.g., Bonferroni Method). In this paper, we demonstrate the quantitative effectiveness of the proposed method in synthetic data experiments, where the true signal and noise are provided as oracles, by showing that it can control type I error (false positive detection probability) at a specified significance level and achieves significantly higher detection power than the Bonferroni correction.

---

> > ### Comment · Reviewer_fh2u · 2024-11-28
> > **Thanks for your responses**
> >
> > Thank you for your response. However, I still have a few remaining concerns:
> >
> > 1. Since the reference image is randomly drawn from the normal examples, the choice of the reference example could influence the results. I strongly recommend conducting a preliminary study to analyze how the results vary with different selections. This would help assess the reliability and robustness of the statistical test.
> >
> > 2. I remain concerned about the lack of quantitative results. Since the method can be tested on synthetic data to produce quantitative results, it seems reasonable to expect similar evaluations on real data. This would provide stronger evidence to support the effectiveness and practicality of the proposed approach.
> >
> > 3. As mentioned in the paper, the method quantifies the statistical significance of the anomaly detection results. Could the method also be used to improve anomaly detection performance, especially in cases where the statistical significance is low?

---

> > > ### Author Response · Authors · 2024-12-02
> > >
> > > > Since the reference image is randomly drawn from the normal examples, the choice of the reference example could influence the results. I strongly recommend conducting a preliminary study to analyze how the results vary with different selections. This would help assess the reliability and robustness of the statistical test.
> > >
> > > Thank you for suggestion. We will conduct additional experiments to address this point and include the results in the revised version. Nonetheless, we would like to emphasize that our method provides p-values that are theoretically guaranteed to control the type I error rate at the desired significance level, regardless of the reference image selected.
> > >
> > > > I remain concerned about the lack of quantitative results. Since the method can be tested on synthetic data to produce quantitative results, it seems reasonable to expect similar evaluations on real data. This would provide stronger evidence to support the effectiveness and practicality of the proposed approach.
> > >
> > > The biomedical image dataset we focus on is highly heterogeneous and diverse, including cases where no apparent abnormal regions are observed even in diseased individuals, and cases where abnormal regions are present even in healthy individuals. The method proposed in this paper does NOT guarantee accurate probabilities for determining whether a case is healthy or diseased. Instead, it assesses the statistical significance of whether the regions identified as abnormal differ from the reference. Therefore, in cases where no apparent abnormal regions exist despite the individual being diseased, a large p-value may result in a determination of no significant difference. Conversely, small p-values may indicate a significant difference even when abnormal regions are observed in healthy individuals. The demonstrations in this paper aim to show that in cases without apparent abnormal regions, the naive p-value tends to be small, leading to false positives, while the selective p-value correctly becomes large, resulting in true negatives. Similarly, in cases with apparent abnormal regions, both the naive p-value and the selective p-value appropriately result in true positives. Nonetheless, we acknowledge that this reviewers' concern is reasonale. In the revised version, we consider how to conduct data preprocessing so that we can perform fair and reasonable quantitative comparisons.
> > >
> > > > As mentioned in the paper, the method quantifies the statistical significance of the anomaly detection results. Could the method also be used to improve anomaly detection performance, especially in cases where the statistical significance is low?
> > >
> > > The objective of our study is to quantify the statistical significance of anomaly detection results obtained using generative models, such as diffusion models. Making decisions based on the p-value derived from our method enables more confident decision-making. For instance, a very small p-value allows for a confident determination of abnormality.

---

### Note · Authors · 2025-01-24

I have read and agree with the venue's withdrawal policy on behalf of myself and my co-authors.